# Inclusion Complexes of Non-Steroidal Anti-Inflammatory Drugs with Cyclodextrins: A Systematic Review

**DOI:** 10.3390/biom11030361

**Published:** 2021-02-27

**Authors:** Gustavo Marinho Miranda, Vitória Ohana Ramos e Santos, Jonatas Reis Bessa, Yanna C. F. Teles, Setondji Cocou Modeste Alexandre Yahouédéhou, Marilda Souza Goncalves, Jaime Ribeiro-Filho

**Affiliations:** 1Laboratory of Investigation in Genetics and Translational Hematology, Gonçalo Moniz Institute (IGM), Oswaldo Cruz Foundation (FIOCRUZ), Salvador, BA 40296-710, Brazil; gustavobiomed116@gmail.com (G.M.M.); vitoria.ohana@outlook.com (V.O.R.eS.); modya006@yahoo.fr (S.C.M.A.Y.); marilda.goncalves@fiocruz.br (M.S.G.); 2Institute of Psychology (IPS), Federal University of Bahia (UFBA), Salvador, BA 40170-055, Brazil; jonatas.reisbessa@gmail.com; 3Agrarian Sciences Center (CCA), Department of Chemistry and Physics (DQF), Federal University of Paraiba (UFPB), Areia, PB 58397-000, Brazil; yanna@cca.ufpb.br

**Keywords:** cyclodextrins, non-steroidal anti-inflammatory drugs, inclusion complexes

## Abstract

Non-steroidal anti-inflammatory drugs (NSAIDs) are one of the most widely used classes of medicines in the treatment of inflammation, fever, and pain. However, evidence has demonstrated that these drugs can induce significant toxicity. In the search for innovative strategies to overcome NSAID-related problems, the incorporation of drugs into cyclodextrins (CDs) has demonstrated promising results. This study aims to review the impact of cyclodextrin incorporation on the biopharmaceutical and pharmacological properties of non-steroidal anti-inflammatory drugs. A systematic search for papers published between 2010 and 2020 was carried out using the Preferred Reporting Items for Systematic Reviews and Meta-Analyses (PRISMA) protocol and the following search terms: “Complexation”; AND “Cyclodextrin”; AND “non-steroidal anti-inflammatory drug”. A total of 24 different NSAIDs, 12 types of CDs, and 60 distinct inclusion complexes were identified, with meloxicam and β-CD appearing in most studies. The results of the present review suggest that CDs are drug delivery systems capable of improving the pharmacological and biopharmaceutical properties of non-steroidal anti-inflammatory drugs.

## 1. Introduction

Inflammation is the body’s response to tissue damage, which aims to restore the integrity of the injured tissue through different mechanisms of induction, regulation, and resolution. Regardless of the aggressor stimulus, this response is essential for the restoration of homeostasis and, therefore, plays an important physiological role [1,2].

While inflammation is essential for the host defense against pathogens, it frequently occurs in the absence of infection, a phenomenon known as sterile inflammation [3,4]. Accordingly, despite the existence of efficient control mechanisms, failures in the resolution of inflammation often occur, which contributes significantly to the pathogenesis of several chronic diseases [5,6]. In these cases, the inflammatory response is the main cause of tissue injury and disease progression, besides contributing to the severity of other comorbidities [7,8].

The classical therapeutic strategies for inflammatory diseases are mostly based on the use of anti-inflammatory drugs that act by either inhibiting the production of pro-inflammatory mediators or preventing the recruitment and activation of leukocytes at the inflammatory site [9,10]. In this context, non-steroidal anti-inflammatory drugs (NSAIDs) are one of the most widely used drug classes in the treatment of inflammation, fever, and pain. The mechanism of action underlying the anti-inflammatory, antipyretic, and analgesic effects of NSAIDs involves the inhibition of cyclooxygenases (COXs), enzymes that are critically responsible for the synthesis of prostaglandins, thromboxanes, and other prostanoids [11,12]. Prostaglandin E_2_ (PGE_2_) is produced in a wide variety of tissues, playing crucial roles in vasodilation and hyperalgesia in addition to interacting with pro-inflammatory cytokines that generate fever [13,14]. Thus, the mechanism of action of NSAIDs results in the inhibition of key clinical signs such as redness, warmth, pain, and swelling [2].

While non-selective NSAIDs inhibit both COX-1 and COX-2 isoforms, selective NSAIDs were developed to preferentially inhibit COX-2, which was expected to potentiate anti-inflammatory effects and reduce adverse effects. However, accumulating evidence has demonstrated that these drugs can induce significant toxicity [15]. Therefore, the development of innovative, safe, and effective anti-inflammatory drugs remains challenging [16].

In the search for innovative strategies to overcome NSAID-related problems, nanotechnology-based formulations have been developed to improve the pharmacokinetic properties of drugs as well as their interaction with their molecular targets. In this context, the incorporation of drugs into cyclodextrins represented a revolution in drug delivery systems [17,18].

Cyclodextrins (CDs) are cyclic oligosaccharides formed by α-D-glucopyranose units linked by α-1.4 bonds, which contain a hydrophobic central cavity and a hydrophilic outer surface [19]. They are classified according to the number of glucopyranose units in α-CD, β-CD, or γ-CD as they present six, seven, and eight units, respectively. They may be presented as derivatives such as hydroxypropyl, methyl, di-methyl, sulfobutyl, and carboxymethyl CDs [20].

Among CDs, β-CD presents the lowest water solubility, and despite presenting some intake restrictions [21], it is the most used CD mainly due to its easy production and lower price [19]. The characteristics of α-CD, β-CD, and γ-CD are summarized in Table 1.

Considering the importance of inclusion complexes in drug development, this study aims to review the impact of cyclodextrin incorporation on the biopharmaceutical and pharmacological properties of non-steroidal anti-inflammatory drugs.

## 2. Methods 

The present study is a systematic review conducted in four scientific databases (Pubmed, Medline, Scopus, and EMBASE) according to the Preferred Reporting Items for Systematic Reviews and Meta-Analyses (PRISMA) protocol [22], using the following search terms: “Complexation”; AND “Cyclodextrin”; AND “Non-steroidal anti-inflammatory drug”. 

This search included all original articles addressing the impact of CD complexation on the biopharmaceutical and pharmacological effects of NSAIDs, published in the last decade (from 2010/01/01 to 2020/02/05, followed date of this study). The following inclusion criteria were adopted: (1) articles investigating inclusion complexes containing at least one CD and one NSAID in the same formulation; (2) articles analyzing the impact of the complexation on parameters directly related to the biological effects of the complexed NSAID; (3) articles with an analysis conducted in vivo, in vitro, ex vivo, in silico, or during clinical research. The exclusion criteria were the following: (1) studies demonstrating physicochemical characterization without pharmacological correlations; (2) studies investigating drugs with an NSAID-like mechanism of action without approval by drug regulatory agencies. Articles randomly found during the theoretical reference search which met the inclusion criteria were also included in the results.

Following the search on the selected databases and considering the randomly found articles, a total of 644 studies (Figure 1) were used for abstract reading, after which 553 articles were excluded as they did not meet the inclusion criteria. After full-text reading of the manuscripts, another 11 articles were excluded, totaling 80 articles selected to compose this review. The authors, type of inclusion complex, and main findings of these articles are summarized in Table 2. The Jeffrey Amazing Statistics Program (JASP) software version 0.9.2 for MacBook Pro 2010 [23] (Jasp Team, 2020) was used to determine the frequency of inclusion complexes, cyclodextrins, and NSAIDs reported in the studies. The data are expressed in pie charts (Figure 2) and the descriptive analysis is shown in the Appendix A.

## 3. Results

A detailed analysis of the studies selected in the present review is shown in Table 2, while the analysis of frequency represented in pie charts is shown in Figure 2, and the descriptive analysis is detailed in the Appendix A. We identified a total of 24 different NSAIDs used in inclusion complexes with cyclodextrins, including meloxicam (n = 11; 12.94%), diclofenac (n = 8; 9.41%), flurbiprofen (n = 8; 9.41%), ibuprofen (n = 7; 8.23%), piroxicam (n = 7; 8.23%), aceclofenac (n = 6; 7.05%), and oxaprozin (n = 6; 7.05%) as the most frequently complexed drugs. Regarding the type of CD, from a total of 12 different molecules, β-CD (40%) and HP-β-CD (34.78%) were the cyclodextrins most used in the obtention of inclusion complexes with NSAIDs. According to the present frequency analysis, 60 distinct inclusion complexes were obtained and studied from the combination of the NSAIDs and CDs described above, including meloxicam/β-CD (n = 9; 7.82 %), piroxicam/β-CD (n = 7; 6.08), flurbiprofen/HP-β-CD (n = 6; 5.21%), ibuprofen/β-CD (n = 4; 3.47%), oxaprozin/Rme-β-CD (n = 4; 3.47%), and piroxicam/HP-β-CD (n = 4; 3.47%) as the most frequent inclusion complexes in the selected studies.

## 4. Discussion

In a worldwide context, significant concern has arisen regarding the efficacy and safety of the currently available anti-inflammatory drugs [103]. There have been specific conditions in which they are either not effective or can cause significant side effects [104,105]. Therefore, strategies aiming to improve the safety and efficacy of NSAIDs, such as CD preparations, have a cornerstone impact on anti-inflammatory therapy.

Consistent evidence has demonstrated that cyclodextrins can form complexes with both organic and inorganic compounds and, therefore, have become widely used in the food, pharmaceutical, and cosmetic industries. These molecules can be used as functional excipients able to prevent volatility, increase stability, solubility, and membrane permeation, and improve organoleptic characteristics of many molecules [106,107]. Importantly, evidence has demonstrated that CDs are promising drug delivery systems for several drugs, improving their physicochemical and biopharmaceutical properties, increasing their bioavailability, and reducing toxicity, which has a significant impact on the pharmacological effects of both synthetic and naturally occurring compounds [108]. 

The three-dimensional conformation of glucopyranose units in the CD macrocyclic structures places the hydrophobic carbon backbones facing the inner part of the cone, providing the hydrophobic characteristic of the cavity. This structural arrangement determines the application of CDs as host structures to form inclusion complexes with diverse poorly water-soluble molecules such as NSAIDs [109]. Additionally, structural changes in the CD structure by the substitution of chemical groups (such as acetyl, hydroxypropyl, dimethyl, and sulphate) have a significant impact on solubility and drug-release properties of the delivery system [110]. 

In this study, β-CD was the most frequently used native cyclodextrin, while hydroxypropyl β-CD was the principal synthetic CD derivative addressed in the manuscripts, which is probably due to the easy production and low cost of this naturally occurring molecule. However, compared to α- and γ-CDs, β-CD has the lowest solubility, and its acceptable daily intake is up to a dose of 5 mg/kg/day [21]. Thus, although CDs are generally recognized as safe, exposure to higher doses of β-CD can lead to toxicity, as highlighted in the study by Kontogiannidou et al. [96], which attributed the loss of superficial cell layers of porcine buccal mucosa to β-CD and its synthetic derivatives. According to Ishiguro et al. [50], the cytotoxicity (hemolytic activity) of flurbiprofen-2-HB-CD inclusion complexes with different degrees of substitution depends on the degree of substitution. Moreover, consistent evidence has demonstrated the safety of the use of β-CD-based formulations both in vitro [51] and in vivo [36]. Importantly, a study by Rescifina et al. [34] in 2019 demonstrated that the complexation of celecoxib with SBE-β-CD significantly increased the cytotoxic activity of this NSAID, in addition to potentiating the cytotoxicity of gemcitabine, demonstrating enhanced anti-cancer activity.

The development of pharmacological effects depends on adequate drug concentration reaching the aimed tissue. Plasma and other aqueous fluids perfuse most tissues in the body; thus, solubilization in aqueous fluids is often required for drug effectiveness [111]. In contrast, to be absorbed from the application site to the blood or to other tissues, the active compound needs to leave the aqueous phase and permeate lipidic membranes [112] (Lennernas, 2014), and therefore, a balance between lipophilicity and hydrophilicity is desirable for a drug candidate [113]. Thereby, solubility and permeability are considered as key aspects to achieve adequate drug bioavailability and pharmacological effect [114]. 

Following this principle, several studies included in the present review used cyclodextrins (especially β-CD and HP-β-CD) to improve the water solubility and increase the bioavailability of NSAIDs such as aceclofenac [29], meloxicam [80], flurbiprofen [54], diclofenac [37], and lornoxicam. It is worth mentioning that increased solubility was associated with improved bioavailability [37,54] and anti-inflammatory activity [70]. In general, NSAIDs are acidic lipophilic compounds characterized by the presence of an aromatic ring bearing an acidic moiety and can differ in their lipophilicities based on their aryl structure and substituents. For instance, the Log P (octanol–water partition coefficient) values for sodium diclofenac, ibuprofen, piroxicam, and paracetamol are 4.51, 3.97, 1.8, and 0.31, respectively [115]. Sodium diclofenac, ibuprofen, and piroxicam are classified as class II drugs and paracetamol as class III according to the Biopharmaceutics Classification System [116,117]. Considering that most NSAID preparations aim for systemic distribution, the development of pharmaceutical formulations based on CD inclusion complexes may be a promising strategy to reverse their low solubility [117,118]. 

The local or topical administration of drugs in the skin or other membranes is an interesting alternative to the oral route, decreasing systemic side effects and avoiding first-pass metabolism [119]. Topical and transdermal drug administration may show advantages compared to oral administration, especially when a regional effect is expected due the possibility of drug application close to the action site, e.g., deeper skin layers, muscles, and blood vessels [120]. In this context, many studies have investigated the effect of CD complexation (in both binary and ternary systems) on the permeation properties of NSAIDs such as aceclofenac [28], diclofenac [40,41], celecoxib [19], and nepafenac [86]. Regarding formulations designed for ocular administration, aqueous solubility in biological fluid as well as lipophilicity to ensure the membrane permeation are also required characteristics to achieved pharmacological effects [121]. Abdelkader and colleagues [40] showed that α-CD, β-CD, γ-CD, and HP-β-CD had little effect on the ocular permeability of diclofenac, although the toxicity of this drug was significantly reduced by association with γ-CD and HP-β-CD. On the other hand, celecoxib/γ-CD/Rme-β-CD [19] and nepafenac/HP-β-CD [86] showed excellent corneal permeability, resulting in increased drug concentration in the cornea, sclera, and retina, demonstrating the relevance of cyclodextrins as drug delivery systems in ocular formulations.

Cyclodextrin-complexed drugs have demonstrated improved pharmacokinetics aspects, resulting in greater efficacy and safety [108]. In a study by Dahiya and collaborators [25], the incorporation of HP-β-cyclodextrin provided a more rapid onset of pharmacological effects of aceclofenac in comparison to the market formulation and pure drug. Importantly, a clinical study using a diclofenac-HP-β-CD formulation in patients with mild or moderate renal insufficiency or mild hepatic impairment indicated its safe use without the requirement of dose adjustment [39]. Moreover, evidence has indicated that β-CD, HP-β-CD, and SBE-β-CD can be successfully used to improve the pharmacokinetic profile of flurbiprofen, indicating that they are promising delivery systems for poorly soluble drugs [52,55]. An interesting approach has been investigated by Hartlieb et al. (2017). A pharmaceutical preparation of ibuprofen-MOF-CD exhibited similar in vivo bioavailability and uptake in blood plasma as pure ibuprofen. However, the complexed samples showed a significant increase in the blood half-life of ibuprofen when compared to the pure drug, pointing out that the pharmacokinetic aspect improvement may change according to the pharmaceutical preparation. Their results suggested that ibuprofen-MOF-CD may be an effective delivery vehicle able to produce extended analgesic effects. Another interesting use of CDs to avoid drug passage through the blood–brain barrier has been described by Wang et al. [122]. 

To evaluate the effectiveness of cyclodextrins in improving the pharmacological effects of NSAIDs, several studies investigated the anti-inflammatory and analgesic properties of different NSAID–CD inclusion complexes, as these drugs have fundamental importance in the management of inflammation and pain. On the other hand, these drugs can present significant side effects, such as increased cardiovascular risk and stimulation of gastric ulcer formation [121]. Grecu and collaborators [68] demonstrated that ketoprofen/β-CD complex presented a stronger anti-inflammatory activity than ketoprofen pure in rat models of paw edema and peritonitis. Similar findings were obtained by Auda [88], Alshehri et al. [123], and Ammar et al. [70] using inclusion complexes of nimesulide-Me-β-CD, flufenamic acid-β-CD, and lornoxicam-β-CD/lornoxicam-HP-β-CD, respectively. The impact of CD incorporation on the anti-inflammatory effects of NSAIDs has been further demonstrated in vitro and was associated with inhibition of inflammatory mediator production [32,83].

According to Singh et al. [47], etoricoxib, a selective COX-2 inhibitor, presented increased solubility and enhanced analgesic activity upon complexation with β-CD and HP-β-CD, exhibiting maximum analgesic effects when complexed to HP-β-CD. The incorporation of meloxicam [77] or aceclofenac [30] to β-CD was found to significantly improve the biopharmaceutical properties and potentiate the analgesic and anti-inflammatory effects of these NSAIDs in vivo. Accordingly, a clinical study demonstrated that piroxicam had its analgesic activity potentiated by incorporation with β-CD [99]. Furthermore, evidence has indicated that this complexation may result in significantly reduced ulcerogenic potential [95]. This finding is corroborated with studies addressing the ulcerogenic potential of NSAIDs complexed with cyclodextrins, which showed that complexation with either β-CD or HP-β-CD significantly reduced gastric ulcer formation in rats treated with indomethacin or piroxicam, as shown in Table 2. Additionally, a clinical study conducted by Gan et al. [38] demonstrated the cardiovascular safety of intravenous HP-β-CD-diclofenac. On the other hand, β-CD significantly enhanced the solubility of sulindac but had no protective effect against gastric ulcer formation in rats [101]. The authors discuss that the reported non-effect is related to sulindac itself, which is characterized by its greater ability for gastrointestinal adhesion and high accumulation that can induce gastric ulcer [101].

Cyclodextrin-based inclusion complexes have proven their usefulness to improve the stability [124] and taste of oral pharmaceutical forms, resulting in improved patient compliance in addition to contributing to the biopharmaceutical properties of oral drugs [107]. Accordingly, this study showed that incorporation of cyclodextrins (such as α-, β-, γ-, and HP-β-CD) had a positive impact on the stability of NSAIDs [24,28,102] as well as demonstrated the potential to be used as taste-masking excipients [26,42,69,72]. 

The use of CDs can be limited depending on the drug structure and the CD structure and size. The guest molecules need to interact by non-covalent bonding and fit totally or partially within the CD cavity. The main aspects to ensure proper complexation are related to the chemical structure and physicochemical properties of the guest and host molecules [31,45]. Several studies have used NSAID–CD inclusion complexes as part of ternary and quaternary systems in association with a wide variety of compounds and nanoparticles (e.g., liposomes, chitosan, L-arginine, and poly (lactic-co-glycolic acid) (PLGA)) to achieve additional improvement of the physicochemical, biopharmaceutical, and pharmacological properties. It has been shown that water-soluble polymers such as PLGA, polyethylene glycol (PEG), and HP-methylcellulose, among others, can reduce CD mobility and increase the complex solubility [31,43,45,48,51,61,82,90,92]. 

Sherje et al. achieved a relevant increase in the solubility of etodolac using a ternary system with HP-β-CD and L-arginine. It has been described that basic amino acids such as L-arginine can complex with the drug and the CD by hydrogen bonding, electrostatic interactions, and salt formation. The amphiphilic characteristic of L-arginine allows the structure to interact its hydrophobic region with the hydrophobic portion of HP-β-CD, resulting in a supramolecular ternary complex [45,71]. The findings of the present systematic review indicate that ternary and quaternary systems may contribute to the effectiveness of some NSAID–CD complexes by promoting better complexation, solubility, and a more controlled drug release.

## 5. Conclusions

The results from this review suggest that cyclodextrins, including 2-HB-CD, DiMe-β-CD, EPI-CM-β-CD, EPI-β-CD, HP-β-CD, Me-β-CD, Rme-β-CD, SBE-β-CD, TA-β-CD, α-CD, β-CD, and γ-CD, can be successfully employed in the obtention of inclusion complexes with NSAIDs such as meloxicam, diclofenac, and flurbiprofen, the most frequently complexed drugs.

While the effectiveness of different cyclodextrins in improving the biopharmaceutical and pharmacological properties of NSAIDs depends on both the complexed drug and the type of CD, an overall analysis of the studies included in the present review showed that drug characteristics, including solubility, stability, taste, toxicity, and bioavailability, in addition to the expected anti-inflammatory and analgesic activity were found to be improved upon complexation with molecules such as β-CD and HP-β-CD, the most frequently used CDs.

In conclusion, the findings of the present systematic review suggest that cyclodextrins are promising drug delivery systems capable of improving the pharmacological and biopharmaceutical properties of non-steroidal anti-inflammatory drugs.

## Figures and Tables

**Figure 1 biomolecules-11-00361-f001:**
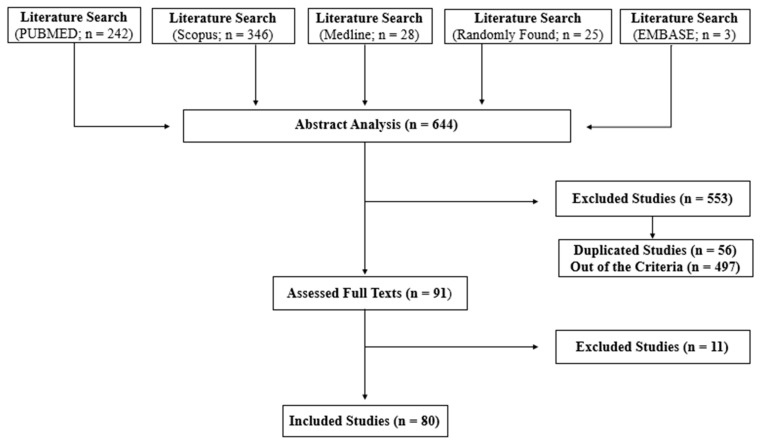
Schematic diagram showing the methodological strategy adopted in this systematic review.

**Figure 2 biomolecules-11-00361-f002:**
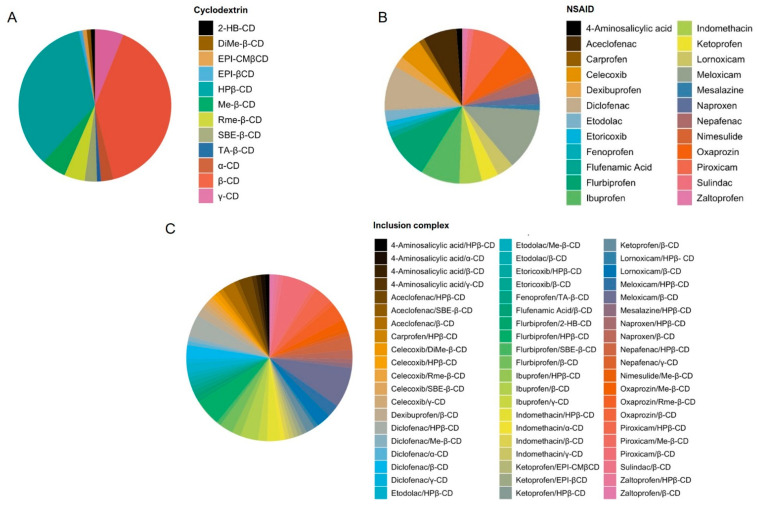
Pie chart plots of frequency analysis. (A) Cyclodextrins; (B) non-steroidal anti-inflammatory drugs (NSAIDs); and (C) inclusion complexes. These data were analyzed using the Jeffrey Amazing Statistics Program (JASP) software version 0.9.2.

**Table 1 biomolecules-11-00361-t001:** Physical and chemical properties of α-, β- and γ-cyclodextrin (CD).

Property	α-CD	β-CD	γ-CD
Glucopyranose units	6	7	8
Molecular weight (Da)	972	1135	1297
Solubility in water (mg/mL)	145	18.5	232
Inner diameter (Å)	5.7	7.8	9.5
Safety (orally)	High *	Acceptable daily intake of 5 mg/kg/day	High *
Chemical structure	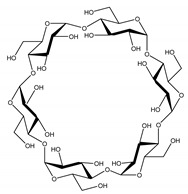	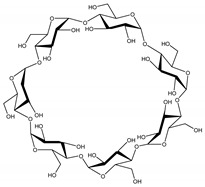	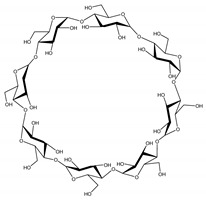

* According to the Regulatory Status of Cyclodextrins, they can be taken without restrictions [21].

**Table 2 biomolecules-11-00361-t002:** Summary of the studies addressing the impact of cyclodextrins on the biopharmaceutical and pharmacological properties of non-steroidal anti-inflammatory drugs.

Authors	Inclusion Complex	Study Design	Main Findings
Lahiani-Skiba et al., 2011 [24]	4-Aminosalicylic acid/α-CD/β-CD/γ-CD/HP-β-CD	Investigated the effect of cyclodextrins on the solubility and chemical stability of 4-Aminosalicylic acid (in vitro).	Complexation with different CDs increased drug solubility, while γ-CD and HP-β-CD significantly increased drug stability 4-fold.
Dahiya et al., 2015 [25]	Aceclofenac/HP-β-CD	Investigated the pharmacokinetic parameters of immediate release aceclofenac tablets (in vitro and in vivo).	Tablets prepared with HP-β-CD provided a more rapid onset of pharmacological effects in comparison to the market formulation and pure drug.
Kasliwal et al., 2011 [26]	Aceclofenac/HP-β-CD	Formulated oral dispersible rapid-disintegration tablet to mask the bitter taste of aceclofenac (in vivo and in vitro).	The formulated tablets tested in humans revealed considerable taste masking and rapid disintegration.
Li et al., 2014 [27]	Aceclofenac/SBE-β-CD	Studied the dissolution profile of the inclusion complex and the controlled-release property of the drug (in vitro).	The formulation enhanced the dissolution profile of aceclofenac and promoted a controlled release of the drug.
Sharma et al., 2016 [28]	Aceclofenac/β-CD/liposomes	Evaluated the improvement in stability and transdermal delivery to the inflammatory sites in osteoarthritis (ex vivo).	Achieved better stability permeation and enhanced skin bioavailability of the drug to epidermis and dermis.
Samal et al., 2012 [29]	Aceclofenac/β-CD	Evaluated the solubility and dissolution rate of aceclofenac in complex with β-CD (in vitro).	The dissolution rate of aceclofenac was significantly higher compared to pure aceclofenac.
Ranpise et al., 2010 [30]	Aceclofenac/β-CD/HP-β-CD	Developed formulations based on inclusion complexes of aceclofenac with β-CD and HP-β-CD and evaluated their anti-inflammatory and analgesic effects (in vivo).	Spray-dried aceclofenac complexed with β-CD presented increased solubility. The formulation presented stronger analgesic and anti-inflammatory effects when compared to the uncomplex drug.
Parra et al., 2016 [31]	Carprofen/HP-β-CD/PLGA nanoparticles	Characterized and analyzed the permeability (ex vivo), biomechanical properties, skin irritation, and anti-inflammatory effects (in vivo) of the inclusion complex.	The complex presented high membrane permeability and induced no evident skin irritation. Topically administered nanoparticles demonstrated strong anti-inflammatory activity.
Cannavà et al., 2013 [32]	Celecoxib/PLGA/DiMe-β-CD.	Evaluated the influence of the polymeric carriers on the pharmacological activity (ex vivo).	Celecoxib-loaded PLGA/DiMe-β-CD microspheres were more effective than the free drug as an anti-inflammatory agent on human chondrocyte tests.
Mennini et al., 2012 [33]	Celecoxib/HP-β-CD/PVP/Chitosan-Ca^2+^-alginate	Developed a new system for colon delivery of celecoxib for systemic and local therapy (in vitro).	The results demonstrated the effectiveness of the proposed complex loaded in chitosan-Ca^2+^-alginate microspheres for colon delivery.
Rescifina et al., 2019 [34]	Celecoxib/SBE-β-CD	Analyzed the cytotoxicity of celecoxib and its inclusion complex against A549 cell lines and investigated the impact of the combined use of the inclusion complex in the anti-cancer activity of gemcitabine.	Complexation increased the water solubility of celecoxib, which presented limited cytotoxic activity against A549 cells. Complexation significantly increased the cytotoxic activity of this NSAID as well as improved the cytotoxicity of gemcitabine.
Jansook et al., 2018 [19]	Celecoxib/γ-CD/Rme-β-CD	Developed celecoxib eye drop formulations containing CD and evaluated drug bioavailability, mucoadhesion, permeability (ex vivo), and in vitro cytotoxicity to the retina cell line.	Formulations containing ternary celecoxib/Rme-β-CD/hyaluronic acid improved mucoadhesion, permeability through membrane barriers (semipermeable membrane, simulated vitreous humor, and scleral tissues), and cytocompatibility with human RPE cell line.
Khalid et al., 2020 [35]	Dexibuprofen/β-CD	Prepared and investigated polymeric nanosponges of β-CD to enhance the solubility of dexibuprofen (in vitro).	The polymeric complexation resulted in a 6-fold increase in dexibuprofen solubilization with 89% drug release within 30 min.
Khalid et al., 2017 [36]	Dexibuprofen/β-CD	Prepared β-CD hydrogel nanoparticles using 2- Acrylamido-2-methylpropane sulfonic acid and N, N′-Methylene bis(acrylamide). Solubility was investigated in vitro, and acute toxicity was investigated in rats (in vivo).	The β-CD polymer hydrogel nanoparticles presented high solubility and excellent physicochemical characteristics. No evident toxicity was observed, indicating low oral toxicity.
Hamdan et al., 2016 [37]	Diclofenac/HP-β-CD	Evaluated effects of the preparation on drug absorption, bioavailability, and dissolution (in vivo and in vitro).	In vivo experiments on rats showed that HP-β-CD had no statistically significant effect on absorption or bioavailability of diclofenac. However, improvement in its in vitro dissolution by HP-β-CD was observed.
Gan et al., 2016 [38]	Diclofenac/HP-β-CD	Examined the efficacy and cardiovascular safety of intravenous HP-β-CD-diclofenac (clinical study).	The investigation suggested that the postoperative use of HP-β-CD-diclofenac does not present any additional cardiovascular risk over the placebo.
Hamilton et al., 2018 [39]	Diclofenac/HP-β-CD	Evaluated the pharmacokinetics of diclofenac and HP-β-CD in patients with mild or moderate renal insufficiency or mild hepatic impairment (clinical study).	The results suggest that diclofenac-HP-β-CD might be administered to patients with mild or moderate renal insufficiency or mild hepatic impairment without requirement of dose adjustment.
Abdelkader et al., 2018 [40]	Diclofenac/α-CD, β-CD, γ-CD, and HP-β-CD	Prepared inclusion complexes and evaluated for corneal permeation, corneal opacity/permeability, and toxicity (ex vivo).	Reduction in ocular toxicity by 3- to 16-fold and comparable corneal permeability to free diclofenac were recorded using γ-CD and HP-β-CD complexes.
Klaewklod et al., 2015 [41]	Diclofenac/β-CD/PEG/K_2_CO_3_	Developed and evaluated the release (in vitro) and skin permeability (Franz diffusion cell method) of a diclofenac gel formulation containing β-CD and K_2_CO_3_ in PEG.	The release and permeation of diclofenac from the formulation were significantly greater than those from a commercial gel, indicating its usefulness for the topical delivery of drugs.
Lenik et al., 2016 [42]	Diclofenac/HP-β-CD	Evaluated taste-masking efficiency of diclofenac complexed with HP-β-CD and different sweeteners using a potentiometric electronic tongue.	Pronounced taste-masking effects were obtained with cyclodextrin, comparable to those obtained with other sweeteners such as acesulfame potassium and sodium saccharin.
Mora et al., 2010 [43]	Diclofenac/Me-β-CD/MEA	Characterized diclofenac-Me-β-CD inclusion complex in the presence of MEA. Drug absorption was analyzed through excised human skin (ex vivo).	Diclofenac solubility was increased by the addition of Me-β-CD and further improved in the presence of MEA. Permeability through excised human skin was increased upon drug complexation.
Vieira et al., 2013 [44]	Diclofenac-β-CD	Analyzed the stability and colonic delivery of a diclofenac- β-CD conjugate in simulated gastric and small intestinal fluids, and in fecal human material.	The inclusion complex was stable in simulated gastric and small intestinal fluids, efficiently liberating diclofenac in less than 2 h within the human fecal slurry.
Sherje et al., 2017 [45]	Etodolac/HP-β-CD/L-arginine	Obtained, characterized, and evaluated, by computational modeling (in silico), the solubility of etodolac/HP-β-CD in the presence of different auxiliary agents.	More stable inclusion complexes were obtained when L-arginine was used as an auxiliary agent. The ternary complex of etodolac with HP-β-CD-L/Arginine improved the solubility of the NSAID.
Ammar et al., 2013 [46]	Etodolac/β-CD/Me-β-CD/HP-β-CD	Used several CDs to improve the physicochemical properties of etodolac through complexation (in vitro).	Complexation to CDs enhanced the aqueous solubility and dissolution rate of etodolac.
Singh et al., 2011 [47]	Etoricoxib/β-CD/HP-β-CD	Developed, characterized, and evaluated the solubility and in vivo analgesic activity (tail flick and hot plate) of binary systems of etoricoxib with cyclodextrins.	Etoricoxib showed increased solubility and enhanced analgesic activity upon complexation with β-CD and HP-β-CD. Etoricoxib-HP-β-CD complex showed maximum analgesic effect.
Ammar et al., 2017 [48]	Fenoprofen/TA-β-CD	Characterized and investigated (in vivo) the analgesic and anti-inflammatory properties of fenoprofen/TA-β-CD inclusion complex conjugated with other polymers.	Fenoprofen calcium dehydrate/TA-β-CD conjugated with ethyl cellulose polymer showed the most potent and sustained anti-inflammatory and analgesic activities, possibly due to controlled drug release.
Alshehri et al., 2020 [49]	Flufenamic acid/β-CD/Soluplus^®^	Investigated the solubility, drug release profile (in vitro), and in vivo anti-inflammatory activity (carrageenan-induced paw edema in rats) of the inclusion complex.	The solubility and anti-inflammatory effect of flurbiprofen were optimized after incorporation into the inclusion complex.
Ishiguro et al., 2011 [50]	Flurbiprofen-2-HB-CD	Prepared HB-β-CDs with different degrees of substitution and characterized their physicochemical and biological properties. The hemolytic activity was evaluated in vitro.	HB-β-CDs acted as fast-dissolving carrier molecules for poorly water-soluble drugs. The hemolytic activity was variable according to the degree of substitution.
Shinde et al., 2019 [51]	Flurbiprofen/HP- β-CD/N-TMC	Analyzed TMC nanoparticles containing flurbiprofen/HP-β-CD inclusion complex for drug release, mucoadhesion, and irritation potential (in vitro).	TMC nanoparticles offered prolonged release potential for transmucosal ocular delivery of the NSAID. HET-CAM studies demonstrated their safety for ocular use.
Wang et al., 2017 [52]	Flurbiprofen/HP-β-CD	Analyzed the impact of penetration enhancers on permeability (ex vivo) and pharmacokinetics (in vivo) of flurbiprofen formulations.	Flurbiprofen/HP-β-CD percutaneous permeability was significantly accelerated by turpentine. In vivo pharmacokinetic study showed increased Cmax, shortened Tmax, and unchanged bioavailability.
Vega et al., 2013 [53]	Flurbiprofen/HP-β-CD/PEG/PLGA	Prepared formulations and evaluated as skin-controlled delivery systems (in vivo and ex vivo).	Greater anti-inflammatory efficacy compared to the control and stronger efficacy in preparations using HP-β-CD.
Li et al., 2010 [54]	Flurbiprofen/β-CD	Evaluated the effect of a drug solution containing a surfactant and β-CD on the solubility, bioavailability, and pharmacokinetics of flurbiprofen.	Solubility and plasma exposure (both AUC and Cmax) of flurbiprofen in the mixed system were significantly higher compared to formulations containing surfactant alone, or β-CD alone.
Zhang et al., 2015 [55]	Flurbiprofen/β-CD, HP- β-CD and SBE- β-CD/liposomes	Characterized the pharmacokinetic profile (in vivo) of inclusion complexes of the NSAID and β-CD, HP-β-CD, and SBE-β-CD loaded into liposomes.	The delivery systems significantly improved the relative bioavailability of flurbiprofen in Wistar rats, indicating that they are promising delivery systems for poorly soluble drugs.
Zhao et al., 2013 [56]	Flurbiprofen/β-CD	Incorporated flurbiprofen/β-CD into PACA nanoparticles and its oral bioavailability was evaluated in rats (in vivo).	The reduced particle size and increased surface area may have contributed to the enhanced oral bioavailability of flurbiprofen in the new formulation.
Celebioglu et al., 2019 [57]	Ibuprofen/HP-β-CD	Produced and characterized ibuprofen-HP-β-CD nanofibrous webs and investigated their solubility in water and artificial saliva.	Ibuprofen-HP-β-CD nanofibrous webs presented excellent solubility in both water and artificial saliva, indicating that they have the potential to be used as fast-dissolving oral drug delivery systems.
Volkova et al., 2020 [58]	Ibuprofen/β-CD	Prepared and evaluated ibuprofen properties in MOFs based on β-CD and K^+^ (in vitro).	Ibuprofen-β-CD inclusion complex showed improved aqueous solubility.
Rehman et al., 2013 [59]	Ibuprofen/β-CD	Prepared tablets containing variable compositions of polysaccharide matrix and assessed for in vitro drug release in simulated gastrointestinal fluids containing digestive enzymes.	Formulations containing ethyl cellulose and β-CD (1:1) presented better drug release profiles and proved to be the most adequate for drug delivery to the colon.
Marianecci et al., 2013 [60]	Ibuprofen/β-CD	Evaluated the permeation properties of ibuprofen-β-CD loaded into non-ionic surfactant vesicles (NSVs) using a two-compartment diffusion cell (in vitro).	Ibuprofen-β-CD-NSV system exhibited significantly improved in vitro drug permeation properties with respect to those of the plain drug suspension.
Li et al., 2017 [61]	Ibuprofen/γ-CD/MOF/PAA	Prepared and characterized composites of γ-CD-based MOFs and PAA. The in vitro cytotoxicity was evaluated using J774 macrophages.	Microspheres composed of CD-MOF nanocrystals embedded in PAA exhibited sustained drug release and presented reduced cell toxicity, compared with pure drug or the drug-γ-CD complex.
Hartlieb et al., 2017 [62]	Ibuprofen-γ-CD-MOF	Obtained and characterized a pharmaceutical cocrystal of ibuprofen-MOF-CD. Cytotoxicity (in vitro) and pharmacokinetics (in vivo) were determined.	The pharmaceutical cocrystal did not affect viability of the cells, while the in vivo bioavailability was not affected, and the formulation presented 100% longer half-life in blood plasma.
Felton et al., 2014 [63]	Ibuprofen, ketoprofen, naproxen, flurbiprofen/β-CD/HP-β-CD	Characterized the properties of different NSAIDs complexed with β-CD and two HP-β-CD derivatives (in silico).	NSAID solubility was significantly increased with increasing CD concentration of both HP-β-CD derivatives, whereas β-CD complexation caused little increase in NSAID solubility.
Mohamed et al., 2011 [64]	Indomethacin/HP-β-CD	Investigated the influence of CD incorporation on the solubility pharmacologic effect of indomethacin in injured eyes of albino rabbits (in vivo).	The complex demonstrated sufficient solubility and effectiveness in healing the corneal lesion, with improved anti-inflammatory activity.
Ribeiro-Rama et al., 2011 [65]	Indomethacin/HP-β-CD	Investigated the hepatic and renal injuries caused by indomethacin in several formulations including complexes with HP-β-CD (in vivo).	The animals administered with indomethacin in complexed form showed similar hepatic and renal lesions to those administered with the drug in acid or salt forms.
Vranic et al., 2010 [66]	Indomethacin/α-CD/γ-CD	The aim of the study was to compare the dissolution profiles of indomethacin alone and complexed with CDs.	The complexation of indomethacin with α- and γ-cyclodextrins resulted in increased dissolution rates.
Cirri et al., 2012 [67]	Ketoprofen/Epi-β-CD/Epi-CM-β-CD	Developed a topical administration system to improve the solubility of ketoprofen.	Demonstrated the usefulness of this new system, with improvement in the dissolution of the drug.
Grecu et al., 2014 [68]	Ketoprofen/β-CD	Compared the anti-inflammatory effects of orally administered ketoprofen and ketoprofen/β-CD inclusion complex using two models of experimentally induced acute inflammation in rats (paw edema and peritonitis).	The complexation of ketoprofen with β-CD resulted in increased solubility and bioavailability compared with ketoprofen. In both models of inflammation, ketoprofen/β-CD complex presented a stronger anti-inflammatory activity than ketoprofen.
Moutasim et al., 2017 [69]	Lornoxicam/β-CD	Formulated tablets containing lornoxicam-β-CD to increase drug solubility and to mask its bitter taste. Bioequivalence was evaluated in healthy volunteers.	The formulation presented acceptable palatability, enhanced dissolution, and rapid onset of drug action. The pharmacokinetic profile was significantly different from the control tablet.
Ammar et al., 2012 [70]	Lornoxicam/β-CD/HP-β-CD	Analyzed the efficacy of lornoxicam by complexation with cyclodextrins in liquid crystalline gel (ex vivo).	The preparation showed high improvement in drug dissolution, superior anti-inflammatory activity, and low skin permeation, being suitable for topical use.
Bramhane et al., 2011 [71]	Lornoxicam/β-CD/L-arginine	Evaluated the dissolution of lornoxicam molecular inclusion with β-CD alone and in combination with arginine (in vitro).	The complexation of lornoxicam with β-CD and arginine resulted in significantly higher dissolution compared with the uncomplexed drug and binary systems.
Samprasit et al., 2018 [72]	Meloxicam/HP-β-CD	Prepared taste-masked meloxicam oral dissolving films and evaluated in human volunteers. Cytotoxicity was evaluated using human gingival fibroblasts (HGF).	A fast disintegration time of meloxicam was obtained from ethanol system, which presented lower cytotoxicity. The films rapidly dissolved in the mouth and had a less bitter taste than meloxicam.
Samprasit et al., 2014 [73]	Meloxicam/HP-β-CD	Developed and characterized oral disintegrating tablets using the combination of ion exchange resin and CD. Solubility (in vitro) and taste (in vivo) were evaluated.	Meloxicam incorporated in a combination of ion exchange resin and HP-β-CD demonstrated a good taste, rapid disintegration, complete solubility, and significant stability.
Rein et al., 2020 [74]	Meloxicam/β-CD	Investigated an in situ forming system based on meloxicam in β-CD for periodontitis treatment, aiming sustainable drug release at a periodontal pocket (in vitro).	The developed system comprising 40% β-CD transformed into microparticles extended the drug release to 7 days in the locality of the treatment.
Jafar et al., 2017 [75]	Meloxicam-β-CD-TEA	Investigated the properties of a buoyant in situ gel prepared from a ternary complex of meloxicam with β-CD and TEA on solubility, stability, and in vivo anti-inflammatory activity (carrageenan model in mice).	The solubility, stability. and anti-inflammatory activity of meloxicam was successfully increased due to its incorporation in ternary complex with β-CD-TEA, demonstrating improved pharmaceutical and pharmacodynamic properties.
Ainurofiq et al., 2016 [76]	Meloxicam/β-CD	Investigated inclusion complexes of meloxicam/β-CD incorporated into orally disintegrating tablets using a quality by design approach.	Although interactions between meloxicam and β-CD occurred under different complexation methods, best solubility and dissolution rate enhancement was achieved with the spray drying method.
Shende et al., 2015 [77]	Meloxicam/β-CD -based nanosponges	Investigated inclusion complexes of meloxicam and β-CD-based nanosponges on stability, solubility, drug release (in vitro and in vivo) and analgesic and anti-inflammatory effect (in vivo).	Incorporation of meloxicam into β-CD-based nanosponges improved its biopharmaceutical properties and potentiated its analgesic and anti-inflammatory effects.
Radia et al., 2012 [78]	Meloxicam/β-CD.	Investigated the effect of Meloxicam complexation with β-CD on the bioavailability of the drug (in vivo and in vitro).	The oral bioavailability of ML was significantly improved through complexation with β-CD.
Rasool et al., 2011 [79]	Meloxicam/β-CD	Developed and evaluated the properties of a meloxicam transdermal gel in complexation with β-CD.	The dissolution properties of the meloxicam-β-CD complexes were superior when compared to meloxicam alone.
Awasthi et al., 2011 [80]	Meloxicam/β-CD	Investigated the improvement on meloxicam solubility by preparation of its solid dispersion using β-CD blended with various water-soluble polymer carriers (in vitro).	Meloxicam formulations obtained by solid dispersion technique presented significantly enhanced solubility, highlighting β-CD blended with 0.12% (w/w) HP-methylcellulose polymer.
Janovský et al., 2010 [81]	Meloxicam/β-CD	Investigated the influence on analgesic activity and serum levels after meloxicam complexation (in vivo).	Stronger analgesic activity than unmodified meloxicam; Serum levels of meloxicam were significantly higher.
Samprasit et al., 2013 [82]	Meloxicam, Piroxicam/β-CD, HP-β-CD	Prepared and characterized inclusion complexes of NSAIDs with either β-CD or HP-β-CD loaded into an anionic exchange resin.	The solubility of both drugs was found to increase with increasing CD concentration. The presence of cyclodextrin in the loading solution resulted in the improvement of drug loading into the resin.
Tang et al., 2018 [83]	Mesalazine/HP-β-CD/chitosan nanoparticles	Analyzed the cytotoxic and anti-inflammatory effects of mesalazine/HP-β-CD/chitosan nanoparticles on cytokine stimulated HT-29 cells (in vitro).	The complexed drug more strongly inhibited the production of inflammatory mediators such as NO, PGE_2_ and IL-8.
Elkordy et al., 2012 [84]	Naproxen/β-CD, Poloxamer-407	Investigated the effects of β-CD, Poloxamer-407 and sorbitol as drug carriers (in vitro).	Different tested mixtures of the drug with β-CD and Poloxamer-407 showed enhancement in drug release.
Shelley et al., 2018 (a) [85]	Nepafenac/HP-β-CD	Investigated trans-corneal permeation of nepafenac in a HP-β-CD complex (ex vivo).	The formulation showed a significantly higher drug concentration in the cornea with a permeation rate 18-fold higher than the commercial product.
Shelley et al., 2018 (b) [86]	Nepafenac/HP-β-CD.	Evaluated the solubility and transcorneal permeation of a gel containing nepafenac-HP-β-CD, in comparison with the commercial nepafenac (ex vivo).	The formulation increased the solubility and ocular release of nepafenac, resulting in a higher concentration of drug in the cornea, sclera and retina compared to the commercial drug
Lorenzo-Veiga et al., 2020 [87]	Nepafenac/γ-CD/HP-β-CD	Analyzed CD-based formulations with several polymers to efficiently deliver nepafenac topically to the eye (in vitro and ex vivo).	Formulations with methylcellulose and carboxymethylcellulose polymers showed improved solubility, mucoadhesion, permeability and anti-inflammatory potential.
Auda, 2014 [88]	Nimesulide/Me-β-CD	Characterized and evaluated the solubility (in vitro) and the anti-inflammatory effect of the inclusion complex in carrageenan-induced inflammation in rats (in vivo).	The study showed that the inclusion system enhanced the solubility and anti-inflammatory activity of the drug.
Mura et al., 2016 [89]	Oxaprozin/Me-β-CD	Used a combined approach based on drug complexation with methylated β-CD and nanoclays to improve the dissolution of oxaprozin (in vitro).	The complexation and clay nanoencapsulation improved the oxaprozin dissolution properties around 100%.
Maestrelli et al., 2011 [90]	Oxaprozin-Rme-β-D/chitosan/bile components	Investigated the effect of the combined use of Rme-β-CD, chitosan, and different bile components on solubility and permeability of oxaprozin through Caco-2 cells.	Chitosan and bile acids increased the Rme-β-CD solubilizing power. The addition of CS to oxaprozin-Rme-β-CD systems increased drug permeability through Caco-2 cells.
Maestrelli et al., 2017 [91]	Oxaprozin/Rme-β-CD/L-arginine	Investigated a possible enhancement of the Rme-β-CD solubilizing efficacy through a combined approach with arginine in sepiolite nanoclay (SV). Anti-inflammatory activity was assessed using CFA-induced arthritis (in vivo).	The new hybrid nanocomposite was more effective than the respective oxaprozin-Rme-β-CD and oxaprozin-SV systems in improving the drug dissolution properties, as well as the effects on CFA-induced arthritis.
Mennini et al., 2016(a) [92]	Oxaprozin/Rme-β-CD/L-arginine	Evaluated the influence of L-arginine on the properties of an oxaprozin/Rme-β-CD inclusion complex, including stability and solubility.	The ternary system presented improved properties, indicating its potential for enhancing safety, bioavailability, and therapeutic efficacy of NSAIDs such as oxaprozin.
Mennini et al., 2016(b) [93]	Oxaprozin/Rme-β-CD/Arginine	Characterized and evaluated the in vitro drug permeability of the drug complexed with Rme-β-CD and loaded into liposomes or nanostructured lipid carriers (NLCs).	The combined use of Rme-β-CD and either liposomes or NLCs enabled increases in the drug permeability through artificial membranes and excised human skin.
Mura, et al., 2010 [94]	Oxaprozin/β-CD/Me-β-CD/PLGA	Investigated the properties of a formulation based on the combined use of CD and PLGA nanoparticles of oxaprozin (in vitro).	The complex was effective in solubilizing and stabilizing the drug, improving the prolonged-release properties of nanoparticles.
Mishra et al., 2019 [95]	Piroxicam/β-CD/Pectin/chitosan/albumin	Studied β-CD-based formulations with several polymers to optimize the delivery and activity of the drug (in vivo).	The drug in the β-CD polymer conjugate was found to possess enhanced analgesic activity and reduced ulcerogenic potential in rats.
Kontogiannidou et al., 2019 [96]	Piroxicam/β-CD/HP-β-CD/Me-β-CD	Obtained and characterized piroxicam-CD tablet formulations. The in vitro release profile was analyzed in simulated saliva and the drug permeation studies, across porcine buccal mucosa (ex vivo).	The drug release profile as well as the tissue permeability followed this order: Me-β-CD > HP-β-CD > β-CD. Chitosan significantly increased the transport of the drug compared to their free complexes. Loss of superficial cell layers was attributed to the presence of CD.
Bouchal et al., 2015 [97]	Piroxicam/β-CD/HP-β-CD	Compared the dissolution of piroxicam in the absence or presence of CDs.	The drug was faster and completely dissolved in an aqueous medium in the conjugated form compared to the free form.
Skiba et al., 2013 [98]	Piroxicam/β-CD	Compared the pharmacokinetic parameters of piroxicam after an oral administration in rabbits (in vivo).	The complexed piroxicam had faster absorption than the free drug but the excess of CD had a negative impact on the permeability of the drug through the biological membrane.
Keleş et al., 2010 [99]	Piroxicam/β-CD	Investigated the analgesic efficacy and adverse effects of preoperatively administered piroxicam-β-CD (clinical study).	Preemptive administration of piroxicam-β-CD effectively reduced analgesic consumption, and 40 mg of the complexed drug was effective without side effects.
Alsarra et al., 2010 [100]	Piroxicam, indomethacin/β-CD, HP-β-CD	Compared the formation of gastric ulcers in rats treated with isolated and complexed drugs (in vivo).	Complexation with either β-CD or HP-β-CD significantly reduced gastric ulcer formation in rats treated with indomethacin or piroxicam.
Haggag et al., 2016 [101]	Sulindac/β-CD/CAP	Characterized and investigated the ability of different polymers to ameliorate sulindac-induced gastric ulcers in rats (in vivo).	β-CD significantly enhanced the solubility of sulindac but had no protective effect against gastric ulcer formation in rats.
Sherje et al., 2018 [102]	Zaltoprofen/β-CD/HP-β-CD/L-arginine	Investigated formulations containing CDs and L-arginine for drug stability, dissolution, and solubility (in vitro).	The addition of L-arginine to drug-CD system increased stability, dissolution, and solubility of zaltoprofen.

Abbreviations: Cyclodextrin (CD); Hydroxypropyl-β-Cyclodextrin (HP-β-CD); Beta cyclodextrin (β-CD); Sulfobutyl ether-β-cyclodextrin (SBE-β-CD); Alpha cyclodextrin (α-CD); Gamma cyclodextrin (γ-CD); Methyl-β-cyclodextrin (Me-β-CD); Polyethylene glycol (PEG); Poly(lactic-co-glycolic acid) (PLGA); Polyvinylpyrrolidone (PVP); Dimethyl-β-Cyclodextrin (DiMe-β-CD); hydroxypropyl methylcellulose (HP-methylcellulose); β-CD-Epichlorohydrin polymer (EPI-β-CD); Carboxymethylated-β-CD-epichlorohydrin polymer (EPI-CM-β-CD); Soluplus^®^ (supramolecular ternary inclusion complex); Metal–organic frameworks (MOFs); Potassium (K^+^); Non-steroidal anti-inflammatory drug (NSAID); Randomly methylated-β-CD (Rme-β-CD); Meloxicam (ML); Nitric oxide (NO); Prostaglandin E-2 (PGE_2_); Retinal pigment epithelium (RPE); Nanostructured lipid carriers (NLCs); Potassium carbonate (K_2_CO_3)_; Hen’s Egg Test chorioallantoic membrane (HET-CAM); Complete Freund’s adjuvant (CFA)-induced; Maximum concentration (Cmax); Time maximum (Tmax); Area under the curve (AUC); Triethanolamine (TEA); Sepiolite nanoclay (SV); Human gingival fibroblast (HGF); N-trimethyl chitosan (TMC); Polyacrylic acid (PAA); Non-ionic surfactant vesicles (NSVs); Monoethanolamine (MEA); Poly(Alkyl-Cyanocrylate) (PACA); Cellulose acetate phthalate (CAP); Interleukin-8 (IL-8); Triacetyl (TA).

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
