# Peer review of "Inclusion Complexes of Non-Steroidal Anti-Inflammatory Drugs with Cyclodextrins: A Systematic Review"

_biomolecules, 2021, doi:10.3390/biom11030361_

Round 1

Reviewer 1 Report

The manuscript reviews the influence of cyclodextrins to nonsteroidal anti-inflammatory drugs properties. However, the manuscript has several drawbacks:

a) Although the authors describe the review as a systematic search for articles on the topic, there are many articles that are not included in the manuscript. I found many of them using Web of Science. The authors should include those as well.

b) Unsolved problems and possible developments should also be discussed. Please expand Discussion chapter in this manner.

c) Table 1. Describe what “high” means (in Safety row). Instead of “limited to” is better to use “acceptable daily intake” (see DOI: 10.2903/j.efsa.2016.4628 ). The same for line 155.

d) Line 155-156: Please use “generally recognized as safe” (GRAS) instead of “generally considered safe”

e) Line 156: Define “high doses”. High doses of sodium chloride can cause health issues as well and we are not saying NaCl is toxic. Moreover, some CD derivatives could be tolerated even at grams/kg/day (see e.g. DOI: 10.1021/js960213f )

f) There are many typos as well as inconsistency in compound names or abbreviations in the manuscript, which should be corrected.

Author Response

The manuscript reviews the influence of cyclodextrins to nonsteroidal anti-inflammatory drugs properties. However, the manuscript has several drawbacks:

  1. a) Although the authors describe the review as a systematic search for articles on the topic, there are many articles that are not included in the manuscript. I found many of them using Web of Science.

Authors: As suggested, we have double checked our findings and the presented results and selected articles are in fully accordance to the Preferred Reporting Items for Systematic Reviews and Meta-Analyses (PRISMA) protocol used in this manuscript to select the articles. As described in the Methods section, we used the search terms “Complexation”; AND “Cyclodextrin”; AND “Non-steroidal anti-inflammatory drug” and the inclusion and exclusion criteria were then applied. The exclusion criteria were the following: 1) studies demonstrating physicochemical characterization without pharmacological correlations; 2) studies investigating drugs with an NSAID-like mechanism of action without approval by drug regulatory agencies. It is possible that some articles mentioned by the reviewer have been excluded on the basis on those criteria. Any further concerns do not hesitate in contact us. Thank you for reviewing our manuscript.

  1. b) Unsolved problems and possible developments should also be discussed. Please expand Discussion chapter in this manner.

Authors: Among the selected articles some unsolved problems were mentioned by:

  • Haggag et al. (2016): the CD preparation had no protective effect against gastric ulcer formation.
  • and Hartlieb et al. (2017): cocrystal of ibuprofen-MOF-CD did not change in vivo bioavailability, however the formulation presented 100% longer half-life in blood plasma.

These points were better discussed in the text and other considerations have been made in lines 208 and 234.

  1. c) Table 1. Describe what “high” means (in Safety row). Instead of “limited to” is better to use “acceptable daily intake” (see DOI: 10.2903/j.efsa.2016.4628 ). The same for line 155.

Authors: Your suggestions were performed. The information about ‘high’has been added as footnote of Table 1: ‘According to the Regulatory Status of Cyclodextrins they can be taken without restrictions [21]’. Thank you for your suggestion.

  1. d) Line 155-156: Please use “generally recognized as safe” (GRAS) instead of “generally considered safe”

Authors: Your suggestion was performed. Thank you for reviewing our manuscript.

  1. e) Line 156: Define “high doses”. High doses of sodium chloride can cause health issues as well and we are not saying NaCl is toxic. Moreover, some CD derivatives could be tolerated even at grams/kg/day (see e.g. DOI: 10.1021/js960213f )

Authors: The correction was performed. Thank you.

  1. f) There are many typos as well as inconsistency in compound names or abbreviations in the manuscript, which should be corrected.

Authors: Thank you for suggestion. We have checked the abbreviations. They are listed as footnote of Table 2.

Reviewer 2 Report

In the submitted review Authors systematically discuss, or rather present, the studies concerning the formation of host-guest complexes between CDs and NSAID. The study is very clearly presented, its main part (Table 2) can be useful for the readers. My detailed comments are listed below.

Lines 23-31, there are too many details here. This is an abstract so its length should be limited.

Line 51: Why cyclooxygenase with capital „c”?

Line 52, here the information about the isoforms of COX are missing.

Lines 61-62, I wouldn’t use the term „nanotechnology-based” to describe the simple host-guest complexes of CD with NSAID.

Line 71, table should be with capital „T”.

Line 71, what about CD derivatives-methylated, hydroxypropylated and so on. The Authors should at least mention their existence since they are very popular as excipients and host molecules, especially in the pharmaceutical industry which is the main topic of this article.

Table 1, safety, either “High” or “high”.

Figure 1, either “N” or “n”.

Table 2, have you observed any trends (increasing, decreasing or constant) in number of publications published per year?

Lines 138-145, this should be moved to introduction as there is no discussion of results there.

Line 150, this is not an addition, this is substitution of hydroxyl groups.

The real discussion starts in line 152.

Line 152, not “natural” but “native”.

Line 180, here the information about the BCS classification would be useful, including the categorization of NSAID.

Line 195 “have” not “has”

Author Response

In the submitted review Authors systematically discuss, or rather present, the studies concerning the formation of host-guest complexes between CDs and NSAID. The study is very clearly presented, its main part (Table 2) can be useful for the readers. My detailed comments are listed below.

Lines 23-31, there are too many details here. This is an abstract so its length should be limited.

Authors: Your suggestion was performed. Thank you for reviewing our manuscript.

Line 51: Why cyclooxygenase with capital „c”?

Authors: It was a mistake; the correction has been performed. Thank you.

Line 52, here the information about the isoforms of COX are missing.

Authors: The information about the COX-1 and COX-2 isoforms is line 59.

Lines 61-62, I wouldn’t use the term „nanotechnology-based” to describe the simple host-guest complexes of CD with NSAID.

Authors: Your suggestion was performed. Thank you.

Line 71, table should be with capital „T”.

Authors: Your suggestion was performed. Thank you.

Line 71, what about CD derivatives-methylated, hydroxypropylated and so on. The Authors should at least mention their existence since they are very popular as excipients and host molecules, especially in the pharmaceutical industry which is the main topic of this article.

Authors: Your suggestion was performed. Thank you.

Table 1, safety, either “High” or “high”.

Authors: Your suggestion was performed. Thank you.

Figure 1, either “N” or “n”.

Authors: Your suggestion was performed. Thank you.

Table 2, have you observed any trends (increasing, decreasing or constant) in number of publications published per year?

Authors: The number of publication per year is presented in Figure 1 of the Supplementary Material. The mentioned trend was not observed.

Lines 138-145, this should be moved to introduction as there is no discussion of results there.

Authors: Some changes have been made to the first paragraph of the discussion, highlighting the justification for the wide use of CDs. Thank you.

Line 150, this is not an addition, this is substitution of hydroxyl groups.

Authors: The correction was performed. Thank you.

The real discussion starts in line 152. Line 152, not “natural” but “native”.

Authors: The correction was performed. Thank you.

Line 180, here the information about the BCS classification would be useful, including the categorization of NSAID.

Authors: The suggestion has been performed. Thank you.

Line 195 “have” not “has”

Authors: The correction was performed. Thank you.

Reviewer 3 Report

In the introduction, the authors would add some sentences about the chemically modified cyclodextrins (add some recent papers about the use of  modified cyclodextrins, see and cite for exemple: https://doi.org/10.1016/j.ijpharm.2020.119391)

Figure 2 : Some abbreviations are not listed in the legend of the Table 2 (2-HB-CD for exemple). The best solution is to add the abbreviations in the legend of the figure 2. Moreover, the authors use the abbreviations of the cited publications but the reviewer think that some CDs are the same ( Me-b-CD and Rme-b-CD). So, the authors need to improve the study by analyzing the degree of substitution of CDs and clearly show the presence of some CD polymers (EPI-bCD and EPI-CMbCD).

Table 2 : 

-For Lahiana-Skiba et al: it's beta or gamma-CD (main findings)?
- For Cannava et al.: celecoxib is first included in DiMe-b-CD and then in PLGA microspheres ? 
- For Khalid et al, 2020: nanosponges of b-CD is a kind of b-CD polymer, so the authors need to classified this item in a new category (same for Shende et al).
- For Khalid et al, 2017: the authors should add some details about b-CD hydrogel nanoparticle composition 
- Mora et al: What is MEA  ?
-Alshehri et al: what is Soluplus ?
- Shinde et al: what is N-TMC ?
- Li et al, 2017: what is PAA ?
- Hartlieb et al, could you add the composition of MOF ?

Finally, the review would be read by a native english speaker.

Author Response

In the introduction, the authors would add some sentences about the chemically modified cyclodextrins (add some recent papers about the use of  modified cyclodextrins, see and cite for example: https://doi.org/10.1016/j.ijpharm.2020.119391)

Authors: Your suggestion was performed. Information about derivatives has been added in line 71. The manuscript suggested was mentioned in the Discussion, line 219. Thank you.

Figure 2: Some abbreviations are not listed in the legend of the Table 2 (2-HB-CD for exemple). The best solution is to add the abbreviations in the legend of the figure 2. Moreover, the authors use the abbreviations of the cited publications but the reviewer think that some CDs are the same ( Me-b-CD and Rme-b-CD). So, the authors need to improve the study by analyzing the degree of substitution of CDs and clearly show the presence of some CD polymers (EPI-bCD and EPI-CMbCD).

Authors: Thank you. Your suggestions were performed. We have checked and corrected all the abbreviations. They are listed as footnote of Table 2.

Table 2: 

-For Lahiana-Skiba et al: it's beta or gamma-CD (main findings)?

 Authors:It has been clarified in the Table.

- For Cannava et al.: celecoxib is first included in DiMe-b-CD and then in PLGA microspheres ? 

Authors:Yes, the procedure was described in this sequence.

- For Khalid et al, 2020: nanosponges of b-CD is a kind of b-CD polymer, so the authors need to classified this item in a new category (same for Shende et al).

Authors:Thank you. The information about the polymer was added to the text.

- For Khalid et al, 2017: the authors should add some details about b-CD hydrogel nanoparticle composition. 

Authors:Thank you. The information about the gel composition was added to the text.

- Mora et al: What is MEA  ? It was added to the footnote of the Table.
-Alshehri et al: what is Soluplus ? It was added to the footnote of the Table.
- Shinde et al: what is N-TMC ? ? It was added to the footnote of the Table.
- Li et al, 2017: what is PAA ? It was added to the footnote of the Table.
- Hartlieb et al, could you add the composition of MOF ? The information was added to the text.

Finally, the review would be read by a native english speaker.

Authors:Thank you for your considerations. We have double checked the language. All suggestions have been performed.

Reviewer 4 Report

Please find attached a PDF file with my comments/remarks

Author Response

Miranda et al. reviewed the literature on inclusion complexes between cyclodextrins (CDs) and

nonsteroidal anti-inflammatory drugs (NSAIDs), including manuscripts reporting the combination

of at least one CD and one NSAID in the same formulation, in the 2010-2020 time span. The survey systematically includes works studying the effect of the complexation on parameters directly related to the biological effects of the complexed NSAID, excluding studies based on physicochemical characterization without pharmacological correlations. The methodological strategy as well as the selection criteria are clear and well described, the selection of the works is appropriate and the topic matches the audience and scope of Biomolecules. The discussion might be extended to provide the reader not only with a summary of recent publications but also with a more critical evaluation of the field. Therefore, while appreciating the effort devoted in the work and recognizing its relevance, I recommend a revision before the article is considered for publication in Biomolecules.

Major points:

  1. I am aware that the representations of CD chemical structures with planar glucose units is

quite common, yet it is incorrect. Chair conformations should be used instead.

Authors: Your suggestion was performed. Thank you.

  1. As a relevant number of papers have used ternary or quaternary systems with NSAID-CD inclusion complexes and other compounds and nanoparticles, more emphasis should be given on their description. Lines 232-235 read “[…] to achieve additional improvement of the physicochemical, biopharmaceutical and pharmacological properties. The findings of the present systematic review indicate that ternary and quaternary systems may contribute to the effectiveness of NSAID-CD complexes by promoting a more controlled drug release”. It is not fully clear, however, which is the added value of a combined system, which favorable effect comes from the CD-derivative and which limits are overcome with the use of additional molecules/polymers/nanoparticles. Although some explanations is given in table 2 for some references (43, 45, 61 and 90 for instance), a more detailed discussion should be given in the text.

Authors: Your suggestion was performed and we have added the information about the role of polymers and basic amino acids on the complex formation with CDs.  Thank you.

  1. A paragraph highlighting the limits of CD-based drug delivery systems for the delivery of NSAIDs is missing. This is somehow related to the above point, as in some case the addition of a third component improves the performances of the whole system. I encourage a more extended discussion on this point in the discussion and/or conclusion sections, underlining that, despite their beneficial effects, CD-inclusion is still not enough for a proper delivery of some given drugs.

Authors: Your suggestion was performed and information about limits of CDs have been added (see line 250). Thank you.

Other minor points:

  1. “Non-steroidal” (with hyphen) was used as search term, whereas “nonsteroidal” (without hyphen) in the title, abstract (line 17) and text (line 48, etc), please explain the reason behind.

Authors: The correct form is ‘non-steroidal’ and it has been corrected all over the manuscript. Thank you.

  1. A minor discrepancy in the number of papers given as output of the literature search exists

between the description of the methodological strategy (section 2) and its schematic diagram

(Figure 1). Section 2 reads (lines 93-96) “Following the search on the selected databases, a

total of 615 studies were found and used for abstract reading […] During the random literature search, 29 articles were selected and included […]”. Contrarily, in Figure 1 the sum  2 of the papers identified by the literature search in the 4 databases is 242 + 346 + 28 + 3 = 619, and the articles randomly found are 25. Please correct where needed.

Authors: The corrections have been performed in the text. Thank you.

  1. Line 100, pie charts are reported in Fig 2 (not Fig 1), please correct.

Authors: Your suggestion was performed. Thank you.

  1. In the supplementary material there are two entries for the same NSAID: “Diclofenac, frequency 5, 5.882%” and “Dicloflenac frequency 3, 3.529%”. The latter is clearly incorrect, and the two must be summed up. The total actually gives the output reported in the main text (line 109 “diclofenac (n = 8; 9.41%)”).

Authors: The correction was performed. Thank you.

  1. A list of abbreviations is suitably given as a note under Table 2, but abbreviations are not used throughout the Table (and the text). As a few (non-exhaustive) examples at p.2: entry 1 (LahianiSkiba et al., 2011 [24]), “cyclodextrins” in the box “study design” (should be CDs); entry 2 (Dahiya et al., 2015 [25]), “HP-β- cyclodextrin” in the box “Inclusion complex” and “hydroxypropyl-βcyclodextrin” in the box “main finding” (should be HP-β-CD). Please do use abbreviations, this will help in table reading. Also, is there a reason why “Dimethyl-βCyclodextrin” is abbreviated to “DiMe-β-Cyclodextrin”? It should be DiMe-β-CD in my understanding. And please pay attention to typing error (i.e. line 124 Alpha Cyclorextrin, line 127 Nonesteroidal).

Authors: Your suggestion has been performed. Thank you.

  1. Please use the same syntax structure in the column “study design” of Table 2 (verb + object).

For instance this is not the case at p. 3 entry 3 (development …), or at p. 3 entry 5 and p. 4 entry 4 (passive verb), but other occurrences exist.

Authors: Your suggestion was performed. Thank you.

  1. Line 177 please define Log P.

Authors: The information has been added. Thank you.

Reviewer 5 Report

Comments on the manuscript by Miranda et al. entitled “Inclusion complexes of nonsteroidal anti-inflammatory drugs with cyclodextrins: A systematic review”:

The authors report a search conducted in scientific databases (Pubmed, Medline, Scopus, and EMBASE) using the search terms: “Complexation”; AND “Cyclodextrin”; AND “Non-steroidal anti-inflammatory drug”. The main problem is that the performed search does not include the names of the individual NSAIDs and many papers that use the name of the drug instead of the “Non-steroidal anti-inflammatory drug” term are missed. It would be more correct to include the names of the individual drugs (24 different NSAIDs) in the search as well. Is it possible to expand the search to include the names of the individual drugs? And how this will affect the results?

The article is well written, full of content material and beneficial information.

Some minor issues that can be considered for further improvement:

  • In the reference list PRISMA to be referenced as: “Moher, D., Liberati, A., Tetzlaff, J., & Altman, D. G. (2009). Preferred reporting items for systematic reviews and meta-analyses: The PRISMA statement. PLoS Medicine, 6(7), e1000097. https://doi.org/10.1371/journal.pmed.1000097”;
  • 4, line 115: pixocam/β-CD.

Author Response

The authors report a search conducted in scientific databases (Pubmed, Medline, Scopus, and EMBASE) using the search terms: “Complexation”; AND “Cyclodextrin”; AND “Non-steroidal anti-inflammatory drug”. The main problem is that the performed search does not include the names of the individual NSAIDs and many papers that use the name of the drug instead of the “Non-steroidal anti-inflammatory drug” term are missed. It would be more correct to include the names of the individual drugs (24 different NSAIDs) in the search as well. Is it possible to expand the search to include the names of the individual drugs? And how this will affect the results?

Authors: Thank you for your considerations. Our method was designed considering that some NSAID are called by different chemical names, such as paracetamol that is often called acetaminophen. If we would search by any drug name, we would miss papers as well. It is worth mentioning that in a systematic review we design the method, select the search terms, time window and the inclusion and exclusion criteria. Our findings then will be based on those. If we change anything, the results might also change.

As described in the Methods section, our results are in fully accordance to the Preferred Reporting Items for Systematic Reviews and Meta-Analyses (PRISMA) protocol used in this manuscript to select the articles. As described in the Methods section, we used the search terms “Complexation”; AND “Cyclodextrin”; AND “Non-steroidal anti-inflammatory drug” and the inclusion and exclusion criteria were then applied. The exclusion criteria were the following: 1) studies demonstrating physicochemical characterization without pharmacological correlations; 2) studies investigating drugs with an NSAID-like mechanism of action without approval by drug regulatory agencies. It is possible that some articles mentioned by the reviewer have been excluded on the basis on those criteria. Any further concerns do not hesitate in contact us. Thank you for reviewing our manuscript.

 Some minor issues that can be considered for further improvement:

  • In the reference list PRISMA to be referenced as: “Moher, D., Liberati, A., Tetzlaff, J., & Altman, D. G. (2009). Preferred reporting items for systematic reviews and meta-analyses: The PRISMA statement. PLoS Medicine, 6(7), e1000097. https://doi.org/10.1371/journal.pmed.1000097”;
  • 4, line 115: pixocam/β-CD.

Authors: Your suggested correction have been performed. Thank you.

Round 2

Reviewer 1 Report

The manuscript was improved and could be accepted in the present form.

Author Response

Thank You.

Reviewer 2 Report

The Authors have followed my comments, I find the manuscript ready to be published.

Author Response

Thank you.

Reviewer 3 Report

the authors modified the manuscript taking into account the comments of the reviewers

Author Response

Thank You.

Reviewer 4 Report

please see the attached file

Author Response

Point 1. The chemical structure of cyclodextrins in the revised manuscript is exactly the same as the original submission. Please correct.

Reply: The chemical structure of cyclodextrins was updated according to your consideration.

Point 4. There is no reason to use a capital letter for Non-steroidal, please replace with non-steroidal

Reply: Thank you for your comment. “Non-steroidal was replaced with non-steroidal where inappropriate.

Point 5. You changed 615 to 644, but there is still a discrepancy between the text and the figure. In figure 1 the sum of the papers identified by the literature search in the 4 databases is 619 (242 + 346 + 28 + 3), while the articles randomly found are 25. In the text (lines 98-101) a total of 644 is reported (which cannot be as it should not include the randomly found papers) and 29 from random literature search.

Reply: The sentence has been rewritten

Point 8. The abbreviation for Dimethyl- - in Table 2 is still - - . It should be DiMe- -CD. Again, please pay attention to typing error. For instance line 132, it was Alpha Cyclorextrin w it is Alpha cyclodrextrin

Reply: Thank you. All typos were corrected.